# *N*-[1,3-Dialkyl(aryl)-2-oxoimidazolidin-4-ylidene]-aryl(alkyl)sulphonamides as Novel Selective Human Cannabinoid Type 2 Receptor (hCB2R) Ligands; Insights into the Mechanism of Receptor Activation/Deactivation

**DOI:** 10.3390/molecules27238152

**Published:** 2022-11-23

**Authors:** Eleonora Gianquinto, Federica Sodano, Barbara Rolando, Magdalena Kostrzewa, Marco Allarà, Ali Mokhtar Mahmoud, Poulami Kumar, Francesca Spyrakis, Alessia Ligresti, Konstantin Chegaev

**Affiliations:** 1Department of Drug Science and Technology, University of Torino, 10125 Torino, Italy; 2Department of Pharmacy, “Federico II” University of Naples, 80131 Naples, Italy; 3National Research Council of Italy, Institute of Biomolecular Chemistry, 80078 Pozzuoli, Italy

**Keywords:** cannabinoid receptors, hCB2R selective ligands, drug design, in silico simulations, mechanism of receptor activation

## Abstract

Cannabinoid type 1 (hCB1) and type 2 (hCB2) receptors are pleiotropic and crucial targets whose signaling contributes to physiological homeostasis and its restoration after injury. Being predominantly expressed in peripheral tissues, hCB2R represents a safer therapeutic target than hCB1R, which is highly expressed in the brain, where it regulates processes related to cognition, memory, and motor control. The development of hCB2R ligands represents a therapeutic opportunity for treating diseases such as pain, inflammation and cancer. Identifying new selective scaffolds for cannabinoids and determining the structural determinants responsible for agonism and antagonism are priorities in drug design. In this work, a series of *N*-[1,3-dialkyl(aryl)-2-oxoimidazolidin-4-ylidene]-aryl(alkyl)sulfonamides is designed and synthesized and their affinity for human hCB1R and hCB2R is determined. Starting with a scaffold selected from the NIH Psychoactive Drug Screening Program Repository, through a combination of molecular modeling and structure–activity relationship studies, we were able to identify the chemical features leading to finely tuned hCB2R selectivity. In addition, an in silico model capable of predicting the functional activity of hCB2R ligands was proposed and validated. The proposed receptor activation/deactivation model enabled the identification of four pure hCB2R-selective agonists that can be used as a starting point for the development of more potent ligands.

## 1. Introduction

The *Cannabis sativa* plant was first mentioned in a Chinese medicine text almost 5000 years ago. During the 19th century, modern medicine officially accepted the use of cannabis as a drug for its antiemetic, analgesic and anticonvulsant effects [1]. The endocannabinoid system (ECS) is involved in several physiological and pathological processes, such as appetite regulation, peripheral energy metabolism, pain, inflammation, cardiovascular regulation, musculoskeletal disorders and cancer [2,3,4,5,6,7,8].

Although the history of cannabis is almost 50 centuries long, the main active ingredient, Δ^9^-tetrahydrocannabinol (Δ^9^-THC), was isolated only in 1964 [9], and it took another 20 years to identify its biological targets, cannabinoid receptors (CBRs). Indeed, in 1988, the human cannabinoid type 1 receptor (hCB1R) was identified in the brain [10], where it is activated by endogenous molecules (endocannabinoids). At present, two endocannabinoids have been identified: *N*-arachidonoylethanolamine (anandamide) and 2-arachidonoylglycerol [11,12]. Five years after the discovery of hCB1R, the human type 2 receptor (hCB2R) was identified. hCB2R is mainly expressed at a peripheral level [13], in particular in the immune system [14], such as the spleen and thymus, where it modulates immune suppression, apoptosis and cell migration [15,16]. While CBR antagonists and inverse agonists promote osteoclast apoptosis and can be used to prevent bone resorption [17], agonists could be used to treat neurodegenerative disorders, drug abuse/addiction, cardiovascular diseases, in particular, neuroinflammation and neuropathic pain [18,19,20], and also to delay tumour progression [21] and to ameliorate renal fibrosis [22].

In the past 25 years, great effort has been made to study and understand the biological role of the endocannabinoid system and to identify small molecules able to modulate either hCB1R or hCB2R. In particular, a high selectivity towards hCB2R would represent a fundamental property for new drug candidates, having a positive effect in the treatment of inflammatory processes and chronic pain but no psychotropic consequences ensuing from hCB1R activation [18,19,20]. Ligands showing various selectivity towards hCB1R or hCB2R have been developed using different approaches. For example, in one recent study, a huge amount (~60.000) of commercially available compounds was tested using a High Throughput Screening platform, identifying a number of hCB2R ligands [23]. Nevertheless, a rational drug design of selective hCB2R ligands has been hindered for a considerable time. The main reason for this information gap was the lack of CBR experimental structures that could explain the differential ligand profile between hCB1R and hCB2R despite their high sequence similarity [24]. Only recently, in 2016, the hCB1R X-ray structure was published [25], followed three years later by that of hCB2R [26], laying the basis for the drug design of hCB2R-selective compounds.

Here, we present a new series of highly selective hCB2R ligands based on a new scaffold: *N*-[2-oxoimidazolidin-4-ylidene]-4-aryl/alkyl sulphonamide and provide a structure-based predictive model for the easy and rapid identification of their functional activity.

## 2. Results and Discussion

### 2.1. Chemistry

Our work started with a blind throughput screening of a small library of in-house compounds on a set of human receptor proteins, where *N*-[1,3-diethyl-2-oxoimidazolidin-4-ylidene]-4-methylbenzenesulfonamide (**1**, Figure 1) was identified as a weak ligand of hCB2R. The binding affinity was rather low (*K_i_* > 10 µM); thus, we decided to further explore and decorate this scaffold to optimize the affinity and possibly identify hCB2R selective agonists. We initially applied the synthetic route reported in the literature and synthesized a small library of variously substituted 1,3-dialkyl-2-oxoimidazolidin-4-ylidene-benzenesulfonamides [27].

The reaction between 1,3-dialkyl-4,5-dihydroxyimidazolidin-2-ones and the corresponding benzenesulfonamides was carried out in an acid medium in a hydroalcoholic solvent (Figure 1). All the compounds were obtained in low to moderate yields.

The diversity of the series was heavily limited by the commercial availability of arylsulfonamides (**3a**–**d**) and by the synthetic route: only symmetrically substituted 1,3-dialkyl-4,5-dihydroxyimidazolidin-2-ones (**2a**–**b**) could be easily obtained.

Indeed, we did not observe any relevant activity for the synthesized compounds towards hCB2R. We thus changed the synthetic strategy to obtain a more diversified library and applied a combinatorial synthetic approach (Figure 2).

The final compounds were obtained by the reaction of readily available sulfonyl chlorides with 1,3-dialkyl(aryl)-4-iminoimidazolidin-2-ones. The 4-imino-1,3-dialkyl(aryl)imidalidine-2-one scaffold was synthesized by the cyclization of 1,3-dialkyl(aryl)-1-cyanomethylureas, which were obtained by the reaction of aminoacetonitriles with the corresponding isocyanates or with carbamoyl chlorides obtained in situ from amine and phosgene or their precursors. Finally, aminoacetonitriles were synthesized from alkylamines and aldehydes in the presence of cyanide ions. Such a synthetic strategy allowed the modification of almost all possible substituents of the 2-oxoimidazolidin-4-ylidene-sulfonamides scaffold. Taking advantage of a certain structural similarity of our compounds to other selective hCB2R agonists (Figure 2) [28], we started by placing bulky substituents (*t*-Bu) on the 1-N atom of the imidazolidinone ring. The alkyl/aryl group on the 3-N atom of the scaffold and the sulfonamide moiety were then extensively varied to better investigate the structure–activity relationship (SAR) within the series. Finally, a bulkier substituent than adamantyl was introduced at the 1-N atom, with the hope that it would increase the hCB2R affinity of our compounds.

In particular, four series of compounds (**9a**–**h**–**12a**–**h**) were designed and synthesized (Figure 3). As mentioned, bulky and lipophilic substituents were introduced at the 1-N atom (R = *t*-Bu or 1-adamantyl), while the phenyl, cyclopropylmethyl or 2-metoxyethyl group was linked to the 3-N-atom of the 2-oxoimidalidine substructure (R′ = Ph; CH_2_(C_3_H_5_); CH_2_CH_2_OCH_3_). The lipophilicity and geometry of the sulfanilamide substituent were widely varied (R″ = CH_3_; C_6_H_5_; *p*-CH_3_C_6_H_4_; *p*-BrC_6_H_4_; *m*-BrC_6_H_4_; 2,4,6-(CH_3_)_3_C_6_H_2_; 1-Naf; 2-Naf). In order to better understand the SAR of our compounds, the variously substituted compounds **13**–**15** (Figure 3) were obtained following the same synthetic approach (Table 1).

### 2.2. Competition Binding Assay

The binding affinities (*K_i_* values) of the compounds for human recombinant hCB1R and hCB2R were determined as previously described [29], using SR144528 as a reference compound [30], and are reported in Table 1. Interestingly, among the tested compounds, fourteen displayed a significant binding affinity for hCB2R, with *K_i_* spanning almost two orders of magnitude (from 0.06 to 3.45 µM) (Appendix A), and no relevant affinity towards hCB1R, with the only exception of compound **12a**, showing *K_i_* values of 0.28 and 0.21 µM for hCB2R and hCB1R, respectively.

Although the number of compounds with a relevant binding affinity (*K_i_* < 10 µM) is limited, a SAR analysis can be drawn, comparing the percentages of radioactive ligand displacement at the maximum tested concentration. The comparison suggests that a bulky alkyl substituent at one of the nitrogen atoms of the imidazoline ring (R) is necessary for hCB2R selectivity. Indeed, all compounds bearing Me or Et groups (**1**, **4a**–**g**) completely lack selectivity or are even more active on hCB1R. The introduction of a *t*-Bu group generates compounds with remarkable selectivity on hCB2R. Using a less bulky *i*-Pr instead of t-Bu does not influence selectivity but reduces the binding affinity (**13** vs. **10g**) by one order of magnitude. Further increasing the sterical hindrance by introducing an adamantyl substituent (**12a**–**h**) does not improve the selectivity and is tolerated only when combined with specific moieties at R′. Indeed, only **12b** and **12e** show interesting binding constants, while **12c**, **12d**, **12f**, **12g** and **12h** significantly lose affinity towards hCB2R. The second substituent of the imidazolyl ring R′ plays a crucial role in compound affinity. In particular, a small apolar cyclopropylmethyl moiety is associated with good affinity and selectivity towards hCB2R (**10a–h**), with the best ligand of the series being **10g** (*K_i_* of 60 nM). On the contrary, the insertion of a bulkier phenyl in R′ (**9a**–**h**) generates compounds with good selectivity but poor affinity (*K_i_* > 10 μM). Finally, the insertion of a more polar methoxyethyl substituent produces less selective and active compounds (**11a**–**h**). It, thus, seems that the geometry and steric hindrance of the cyclopropyl ring are quite important for ligand binding to hCB2R. Indeed, the substitution with the corresponding “open form”—*i*-Bu drastically reduces the compound affinity (see **10c** vs. **14**). The different location of R and R′ substituents does not seem to significantly influence the affinity, at least in the case of **10b** and **15**. The effect of the substituents on the sulfonilimide portion R″ is far more difficult to rationalize. Quite different binding affinity has been observed for compounds having the same moiety in R″ but a different combination of substituents in R and R′. However, the presence of a bulky substituent seems to be mandatory for good selectivity, while the presence of a methyl group generates compounds that lack selectivity (**10a** and **12a**). The best substituents for a good affinity seem to be the 3-BrPh and 1-Naf groups.

It has been recently reported that minimal variation in the structure can switch the functional activity of hCB2R ligands from antagonism to agonism and vice versa. For instance, it has been shown that the single shift of a methyl substituent from the 1-N to the 2-N atom of a pyrazole ring is able to turn an agonist into an inverse agonist or neutral antagonist [29].

### 2.3. Docking Studies

Taking advantage of the crystal structures of hCB2R, co-crystallized with agonist and antagonist ligands, we decided to investigate a mechanistic model able to predict the functional activity of our newly synthesized compounds. The X-ray structure co-crystallized with the selective agonist AM12033 and the antagonist AM10257 (PDB codes 6KPC [24] and 5ZTY [26], respectively) has been used for the following in silico studies. Both hCBRs present a 7-transmembrane (TM) bundle folding with an intracellular amphipathic helix. The binding site is extremely hydrophobic and, in hCB2R, is lined by Phe87, Phe91, His95, Val113, Thr114, Phe117, Phe183, Ile186, Trp194, Trp258 and Phe281, belonging to helices TM2, TM3, TM5, TM6 and to the extracellular loop 2. As a consequence, ligands mainly establish hydrophobic interactions at the hCB2R binding site, even if the cognate agonist ligand AM12033 can form hydrogen bonds with residues Ser285, Leu182 and Tyr190. The superposition of the agonist- and antagonist-like structures (Figure 4) returns a quite good alignment in the general folding, as expected, and at the binding site level. The main differences can be observed at the level of TM1, TM2, TM6 and TM7, showing a slight displacement in the two states. Particularly interesting is the position of Trp258 (TM6, Figure 4), which is more oriented towards the binding site in the agonist-like state (PDB code 6KPC) while being more open in the antagonist-like one (PDB code 5ZTY). The latter is, indeed, a rare rotamer of Trp258, only observed in muscarine acetylcholine [26,31,32,33] and neurotensin receptors [34], in which it likely constrains the movement of TM6, stabilizing the inactive conformation of the receptors [26]. This peculiar rotamer has been only observed, up to now, in hCB2R and not in hCB1R. This, in combination with other different structural rearrangements, could, in part, explain the specific character of many compounds, likely better fitting one form than the other and having an agonist or antagonist effect [26]. We thus hypothesized that in hCB2R, as in other previously mentioned GPCRs, the agonist/antagonist effect can be associated with the stabilization of Trp258 in one of the two observed conformations and with the consequent stabilization of one of the two receptor states.

To further verify the Trp258 toggle switch hypothesis, we first performed docking studies of published compounds only differing, as previously mentioned, for the location of a methyl substituent [29]. We simulated, in particular, compounds **47**/**53** (N2- and N1-methyl analogues, respectively) and **49**/**51** (2-N- and 1-N-methyl analogues, respectively; Appendix A) in both 6KPC and 5ZTY X-ray structures. As shown in Appendix A, compounds **47** and **49**, bearing the methyl substituent in 2-N and having an antagonist effect, perfectly fit the hCB2R antagonist-like binding site, superposing the adamantane ring with that of the cognate ligand. On the contrary, in the agonist-like conformation, the different conformation of Trp258 forced compounds **47** and **49** to slightly back off (Appendix A). In contrast, 1-N-methyl substituted compounds 51 and 53 were well fit to the agonist-like state (PDB code 6KPC), having enough room even when Trp258 is in the agonist-like conformation (Appendix A). We could, thus, hypothesize that the 2-N-substituted compound can be better accommodated in the antagonist-like conformation, or better, they might shift the conformational equilibrium and induce the receptor to assume the antagonist-like form. We could thus further support the hypothesis that the Trp258 toggling switch is essential for activating downstream signalling in the case of hCB2R [24]. Similarly, Trp258 plays a crucial role in hCB1R activation, where, however, Phe117 is also fundamental to activate the receptor by means of a more complex twin toggle switch mechanism [24].

On this basis, we submitted to docking simulations the compounds from the series showing the highest activity and selectivity. The compounds showing the best fitting at the binding site (in terms of docking scores and interactions) and the clearest behaviour in terms of agonism/antagonism were **10b**, **10e**, **10g**, **12a**, **12b** and **12e**. In particular, **10b**, **10e**, **10g**, only differing at the R″ substituent, demonstrated to well fit the binding site of the agonist-like form (Figure 5a and Appendix A) because of a reduced steric hindrance in the correspondence of Trp258, which can assume the closer conformation. Additionally, compound **12a** (Figure 5b), even after having changed both R and R″, can easily fit the agonist-like state by switching the oxo-imidazolidine scaffold orientation but completely loses selectivity.

On the contrary, compounds **12b** and **12e** show a higher complementarity in the antagonist-like form (Figure 6 and Appendix A), quite well resembling the orientation of the cognate ligand, with the adamantyl substituent very well superimposed and the rest of the ligand completely filling the binding site up to Trp258 (Appendix A).

### 2.4. Functional Activity

The above-mentioned compounds were tested for their functional activity. In agreement with the molecular docking studies, we found that compounds **10b**, **10e**, **10g** and **12a** activated hCB2R with typical agonist behavior by reducing the cAMP levels induced by NKH-477, as expected for a Gi protein-coupled receptor agonist (Figure 7).

We then confirmed whether compounds **12e** and **12b** also act as antagonists, as predicted by the in silico model. Indeed, both compounds did not alter the level of cAMP upon NKH-477 stimulus (not shown). However, when tested in the presence of an EC80 concentration of a hCB2-ligand (4 µM of JWH-133 agonist challenge), compound **12e** was able to fully antagonize the JWH-133-induced inhibition of NKH-477-induced cAMP formation, thus confirming its predicted behavior (Figure 8). Unexpectedly, compound **12b** did not antagonize the agonist challenge. Only a small displacement was observed and only at the highest concentration (Figure 8). We reasoned that the discrepancy with the predictions was mainly due to the chemical–physical features of the molecule and that the lack of antagonism was caused by issues related to compound solubility in the buffer used for the assay. Indeed, the maximum concentration of **12b** that could be reached in the buffer solution used for the functional assay, measured by UV absorption as well as by HPLC, was 2.3 µM.

## 3. Materials and Methods

### 3.1. Chemical Synthesis

All solvents were purified and degassed before use. Chromatographic separation was carried out under pressure on Merck silica gel 60 using flash-column techniques. Reactions were monitored by thin-layer chromatography (TLC) carried out on 0.25 mm silica-gel-coated aluminum plates (60 Merck F_254_). Unless it is specified, all reagents were used as received without further purifications. Dichloromethane was dried over P_2_O_5_ and freshly distilled under nitrogen prior to use. ^1^H and ^13^C NMR spectra were recorded at room temperature on a JEOL ECZ-R 600 instrument at 600 and 150 MHz, respectively, and calibrated using SiMe_4_ as an internal reference. Chemical shifts (δ) are given in parts per million (ppm). The following abbreviations were used to designate the multiplicities: s = singlet, d = doublet, dd = doublet of doublet, t = triplet, and m = multiplet. ESI spectra were recorded on a Micromass Quattro API micro (Waters Corporation, Milford, MA, USA) mass spectrometer. Data were processed using a MassLynxSystem (Waters). The purity of the final compound was determined by analytical HPLC analyses on Merck LiChrospher C18 end-capped column (250 × 4.6 mm ID, 5 µm) using CH_3_CN 0.1% TFA/H_2_O 0.1% TFA as a solvent and the column effluent was monitored using UV as a detector.

#### 3.1.1. General Synthetic Procedure for N-[1,3-Dialkyl-2-oxoimidazolidin-4-ylidene]arylsulfonamides

To the solution of appropriate benzensulfonamide (5.0 mmol) and 4,5-dihydroxy-1,3-dialkylimidazolidin-2-one (5.0 mmol) in methanol (4 mL), few drops of conc. HCl solution were added and the reaction was heated at reflux for half an hour. Then, it was cooled down to r.t. and the product was isolated as described.

N-[1,3-Diethyl-2-oxoimidazolidin-4-ylidene]-4-methylbenzenesulfonamide (**1**): the reaction mixture was cooled in an ice bath, and the precipitate was filtered off and crystallized from EtOH. Yield: 380 mg; 25%. ^1^H-NMR (CDCl_3_) δ: 1.14–1.24 (m, 6H, 2CH_3_), 2.43 (s, 3H, ArCH_3_), 3.47 (q, J^3^_HH_ = 7.3 Hz; 2H, CH_2_), 3.64 (q, J^3^_HH_ = 7.3 Hz; 2H, CH_2_), 4.59 (s, 2H, CH_2_), 7.28–7.33 (m, 2H), 7.82–7.84 (m, 2H) (C_6_H_4_); ^13^C-NMR (CDCl_3_) δ: 12.6, 12.9, 35.8, 37.7, 48.7, 126.5, 129.4, 138.3, 143.4, 154.3, 164.2. MS (ESI^−^) *m*/*z* 308.2 (M-H)^−^.

N-[1,3-Dimethyl-2-oxoimidazolidin-4-ylidene]benzenesulfonamide (**4a**): the reaction mixture was evaporated, and the residue was purified by flash chromatography (eluent PE/EtOAc = 5/5) to give a colorless oil that solidified in the desiccator. The solid was further crystallized from CCl_4_. Yield: 920 mg; 69%. ^1^H-NMR (CDCl_3_) δ: 3.03 (s, 3H, CH_3_), 3.08 (s, 3H, CH_3_), 4.63 (s, 2H, CH_2_), 7.50–7.62 (m, 3H), 7.94–7.97 (m, 2H) (C_6_H_5_); ^13^C-NMR (CDCl_3_) δ: 27.2, 29.9, 51.4, 126.7, 129.0, 132.8, 141.0, 155.0, 164.7. MS (ESI^−^) *m*/*z* 266.2 (M-H)^−^.

N-[1,3-Dimethyl-2-oxoimidazolidin-4-ylidene]-4-methylbenzenesulfonamide (**4b**): the reaction mixture was cooled in an ice bath, and the precipitate was filtered off and crystallized from EtOH. Yield: 420 mg; 30%. ^1^H-NMR (CDCl_3_) δ: 2.43 (s, 3H, ArCH_3_), 3.02 (s, 3H, CH_3_), 3.07 (s, 3H, CH_3_), 4.61 (s, 2H, CH_2_), 7.30–7.32 (m, 2H), 7.81–7.84 (m, 2H) (C_6_H_4_); ^13^C-NMR (CDCl_3_) δ: 21.6, 27.1, 29.9, 51.3, 126.8, 129.5, 138.3, 143.6, 155.1, 164.5. MS (ESI^−^) *m*/*z* 280.3 (M-H)^−^.

4-Chloro-N-[1,3-dimethyl-2-oxoimidazolidin-4-ylidene]benzenesulfonamide (**4c**): the reaction mixture was cooled in an ice bath, and the precipitate was filtered off and crystallized from EtOH. Yield: 420 mg; 28%. ^1^H-NMR (CDCl_3_) δ: 3.02 (s, 3H, CH_3_), 3.07 (s, 3H, CH_3_), 4.60 (s, 2H, CH_2_), 7.47–7.50 (m, 2H), 7.87–7.89 (m, 2H) (C_6_H_4_); ^13^C-NMR (CDCl_3_) δ: 27.1, 29.8, 51.4, 128.1, 129.1, 139.2, 139.8, 154.9, 164.8. MS (ESI^−^) *m*/*z* 300.1/302.1 (M-H)^−^.

4-Bromo-N-[1,3-dimethyl-2-oxoimidazolidin-4-ylidene]benzenesulfonamide (**4d**): the reaction mixture was cooled in an ice bath, and the precipitate was filtered off and crystallized from EtOH. Yield: 480 mg; 28%. ^1^H-NMR (CDCl_3_) δ: 3.04 (s, 3H, CH_3_), 3.08 (s, 3H, CH_3_), 4.62 (s, 2H, CH_2_), 7.65–7.68 (m, 2H), 7.80–7.83 (m, 2H) (C_6_H_4_); ^13^C-NMR (CDCl_3_) δ: 26.9, 29.7, 51.0, 127.9, 128.1, 132.7, 140.0, 154.8, 164.7. MS (ESI^−^) *m*/*z* 346.1/344.1 (M-H)^−^.

N-[1,3-Diethyl-2-oxoimidazolidin-4-ylidene]benzenesulfonamide (**4e**): the reaction mixture was evaporated, and the residue was purified by flash chromatography (eluent PE/Acetone = 8/2) to give a colorless oil that solidified in the desiccator. The solid was further crystallized from *i*-Pr_2_O. Yield: 860 mg; 58%. ^1^H-NMR (CDCl_3_) δ: 1.15–1.24 (m, 6H, 2CH_3_), 3.47 (q, J^3^_HH_ = 7.3 Hz; 2H, CH_2_), 3.65 (q, J^3^_HH_ = 7.3 Hz; 2H, CH_2_), 4.60 (s, 2H, CH_2_), 7.50–7.59 (m, 3H), 7.94–7.97 (m, 2H) (C_6_H_5_); ^13^C-NMR (CDCl_3_) δ: 12.6, 13.0, 35.9, 37.9, 48.8, 126.6, 128.9, 132.7, 141.4, 154.4, 164.4. MS (ESI^−^) *m*/*z* 294.2 (M-H)^−^.

4-Chloro-N-[1,3-diethyl-2-oxoimidazolidin-4-ylidene]benzenesulfonamide (**4f**): the reaction mixture was evaporated, and the residue was purified by flash chromatography (eluent PE/Acetone = 8/2) to give a colorless oil that solidified in the desiccator. The solid was further crystallized from *i*-Pr_2_O. Yield: 460 mg; 28%. ^1^H-NMR (CDCl_3_) δ: 1.15–1.25 (m, 6H, 2CH_3_), 3.47 (q, J^3^_HH_ = 7.3 Hz; 2H, CH_2_), 3.64 (q, J^3^_HH_ = 7.3 Hz; 2H, CH_2_), 4.59 (s, 2H, CH_2_), 7.48–7.51 (m, 2H), 7.87–7.90 (m, 2H) (C_6_H_4_); ^13^C-NMR (CDCl_3_) δ: 12.6, 13.0, 36.0, 37.9, 48.9, 128.1, 129.2, 139.2, 139.9, 154.2, 164.6. MS (ESI^−^) *m*/*z* 328.2/330.3 (M-H)^−^.

4-Bromo-N-[1,3-diethyl-2-oxoimidazolidin-4-ylidene]benzenesulfonamide (**4g**): the reaction mixture was evaporated, and the residue was purified by flash chromatography (eluent PE/Acetone = 85/15) to give a white solid that was further crystallized from *i*-Pr_2_O. Yield: 395 mg; 21%. ^1^H-NMR (CDCl_3_) δ: 1.15–1.25 (m, 6H, 2CH_3_), 3.47 (q, J^3^_HH_ = 7.3 Hz; 2H, CH_2_), 3.64 (q, J^3^_HH_ = 7.3 Hz; 2H, CH_2_), 4.59 (s, 2H, CH_2_), 7.65–7.67 (m, 2H), 7.80–7.83 (m, 2H) (C_6_H_4_); ^13^C-NMR (CDCl_3_) δ: 12.6, 13.0, 36.0, 37.9, 48.9, 127.6, 128.2, 132.2, 140.4, 154.2, 164.6. MS (ESI^−^) *m*/*z* 372.1/374.1 (M-H)^−^.

(*tert*-Butylamino)acetonitrile (**6a**): to the solution of t-BuNH_2_ (12.5 mL, 0.119 mol), KH_2_PO_4_ (17.0 g, 0.125 mol) and KCN (7.70 g, 0.119 mol) in water (200 mL), paraformaldehyde (3.60 g, 0.119 mol) was added in one portion and reaction was mixed at r.t. for 2 h. Then, NaHCO_3_ sat. sol. (100 mL) was added, and the water phase was extracted with Et_2_O (3 × 150 mL). Joined organic extracts were washed with brine, dried and evaporated. The obtained liquid was distilled under reduced pressure to produce a colorless liquid (bp 74–75 °C; 20 mbar). Yield 8.40 g; 63%. ^1^H-NMR (CDCl_3_) δ: 1.16 (s, 9H, t-Bu), 3.54 (s, 2H, CH_2_), ^13^C-NMR (CDCl_3_) δ: 28.8, 31.0, 51.4, 119.8. MS (ESI^+^) *m*/*z* 113.0 (M+H)^+^.

(1-Adamantylamino)acetonitrile (**6b**): the product was obtained following the same procedure, starting from 1-adamantylamine (9.0 g; 0.060 mol). The resulting 1-Adamantylaminoacetonitrile was precipitated from the reaction mixture and collected by vacuum suction, washed with a small amount of cold water and dried. Yield 9.55 g; 84%. ^1^H-NMR (DMSO-d6) δ: 1.51–1.61 (m, 12H, 6CH_2_), 2.01 (m, 3H, 3CH), 2.29 (m, 1H, NH), 3.54 (d, J^3^_HH_ = 6.2 Hz; 2H, CH_2_); ^13^C-NMR (DMSO-d6) δ: 28.8, 28.9, 36.1, 41.6, 50.4, 121.4. MS (ESI^+^) *m*/*z* 191.3 (M+H)^+^.

1-*tert*-Butyl-3-phenyl-4-iminoimidazolidin-2-one (**8a**): to the solution of **6a** (1.5 mL, 12.0 mmol) in Et_2_O (40 mL), cooled in an ice bath, PhNCO (1.1 mL, 10.2 mmol) was added in one portion. The reaction mixture was vigorously stirred for 10 min, when abundant white precipitate formed, blocking magnetic stirring. The reaction mixture was then stirred manually for 5 min; after that, magnetic stirring was restored. The ice bath was removed and the reaction was mixed for an additional 20 min. The precipitate was collected by vacuum suction, washed with a small amount of cold Et_2_O and dried. Then, the obtained solid was placed in MeOH (10 mL) and the obtained mixture was heated until all solids were dissolved; 10 M NaOH solution (0.25 mL) was added, and the reaction was heated at reflux for 30 min. The reaction mixture was diluted with H_2_O (50 mL) and extracted with CH_2_Cl_2_ (3 × 40 mL). Joined organic extracts were washed with brine, dried and evaporated. The obtained oil was purified by flash chromatography (eluent CH_2_Cl_2_/Acetone 8/2) to give the title compound as a white solid. Yield 1.40 g; 60%. ^1^H-NMR (CDCl_3_) δ: 1.48 (s, 9H, t-Bu), 4.22 (s, 2H, CH_2_), 7.29–7.49 (m, 5H, C_6_H_5_); ^13^C-NMR (CDCl_3_) δ: 27.8, 47.7, 54.2, 127.4, 128.3, 129.6, 132.1, 155.4, 159.1. MS (ESI^+^) *m*/*z* 232.3 (M+H)^+^.

1-*tert*-Butyl-3-(cyclopropylmethyl)-4-iminoimidazolidin-2-one (**8b**): to the solution of **6a** (2.0 mL, 15.6 mmol) in dry CH_2_Cl_2_ (120 mL), Et_3_N (2.3 mL, 16.5 mmol) was added, followed by triphosgene (1.55 g, 5.2 mmol). After 1.5 h, the reaction was completed (TLC control with eluent CH_2_Cl_2_/MeOH 99/1). An excess of Et_3_N (2.3 mL, 16.5 mmol) was added, followed by cyclopropylmethylamine (1.35 mL, 15.6 mmol). After 1 h, the organic phase was washed with HCl (3 × 50 mL), H_2_O (100 mL), NaHCO_3_ sat. sol. (50 mL) and brine. Then, it was dried and evaporated to give a colorless oil. The oil was dissolved in MeOH (20 mL), and 0.5 mL of NaOH 10 M was added under magnetic stirring. After 15 min, the reaction was completed (TLC control with eluent CH_2_Cl_2_/MeOH). The reaction mixture was diluted with H_2_O (50 mL) and extracted with CH_2_Cl_2_ (3 × 30 mL). Joined organic extracts were washed with brine, dried and evaporated. The obtained oil was purified by flash chromatography (eluent CH_2_Cl_2_/MeOH 97/3) to give a title compound that solidified upon standing in the desiccator. Yield 2.25 g; 69%. ^1^H-NMR (CDCl_3_) δ: 0.32–0.34 (m, 2H, CH_2_), 0.46–0.48 (m, 2H, CH_2_), 1.14–1.16 (m, 1H, CH), 3.37 (d, J^3^_HH_ = 6.9 Hz; 2H, CH_2_), 1.40 (s, 9H, t-Bu), 4.00 (s, 2H, CH_2_); ^13^C-NMR (CDCl_3_) δ: 3.7, 9.6, 27.8, 43.2, 47.4, 53.7, 156.8, 161.0. MS (ESI^+^) *m*/*z* 210.2 (M+H)^+^.

1-*tert*-butyl-4-imino-3-(2-methoxyethyl)imidazolidin-2-one (**8c**): the compound was obtained following the same procedure used to synthesize **8b,** starting from 2-methoxyethylamine (1.50 mL; 17.3 mmol). We obtained a yellow oil that was unstable at r.t., so it was used immediately without further purification. Yield 2.45 g; 66%.

1-Adamant-1-yl-3-(cyclopropylmethyl)-4-iminoimidazolidin-2-one (**8d**): the compound was obtained following the same procedure used to synthesize **8c**, starting from 1-adamantylaminoacetonitrile (1.90 g; 0.01 mol). The product precipitated from the reaction mixture was collected by vacuum suction, washed with a small amount of cold MeOH and dried. Yield 1.85 g; 64%. ^1^H-NMR (DMSO-d6) δ: 0.24–0.25 (m, 2H, CH_2_), 0.36–0.38 (m, 2H, CH_2_), 1.07 (m, 1H, CH), 1.62 (m, 6H), 2.03 (m, 9H) (Ad), 3.16 (d, J^3^_HH_ = 5.2 Hz; 2H, CH_2_), 4.02 (s, 2H, CH_2_), 7.70 (br.s, 1H, NH); ^13^C-NMR (DMSO-d6) δ: 3.4, 9.7, 28.9, 35.8, 39.3, 42.1, 45.8, 48.6, 53.5. MS (ESI^+^) *m*/*z* 288.3 (M+H)^+^.

#### 3.1.2. General Synthetic Procedure for N-[1-*tert*-Butyl-2-oxo-3-phenylimidazolidin-4-ylidene]sulphonamides (**9a**–**h**)

To the solution of **8a** (490 mg, 2.12 mmol) and Et_3_N (0.33 mL, 2.37 mmol) in dry CH_2_Cl_2_ (15 mL), cooled in an ice–salt bath, the corresponding sulfonyl chloride (2.10 mmol) was added in one portion. After 10 min, the ice bath was removed, and the reaction mixture was stirred at r.t. until completed (TLC control). Then, the reaction mixture was diluted with CH_2_Cl_2_ (50 mL), and the organic phase was washed with HCl 1N sol. (20 mL), H_2_O (20 mL), NaHCO_3_ sat. sol. (20 mL) and brine. The organic phase was dried, and the solvent evaporated under reduced pressure. The obtained solid was purified as described.

N-[1-*tert*-Butyl-2-oxo-3-phenylimidazolidin-4-ylidene]-methanesulfonamide (**9a**): the solid was purified by crystallization from hot EtOH to give the title compound as a white solid. Yield 140 mg; 22%. ^1^H-NMR (CDCl_3_) δ: 1.50 (s, 9H, t-Bu), 3.02 (s, 3H, CH_3_), 4.75 (s, 2H, CH_2_), 7.35–7.49 (m, 5H, C_6_H_5_), ^13^C-NMR (CDCl_3_) δ: 27.8, 42.4, 47.3, 55.2, 127.1, 128.8, 128.9, 131.7, 153.1, 163.6; MS (ESI^−^) *m*/*z* 308.3 (M-H)^−^.

N-[1-*tert*-Butyl-2-oxo-3-phenylimidazolidin-4-ylidene]benzenesulfonamide (**9b**): the title compound was purified by crystallization from hot EtOH. White solid. Yield 420 mg; 54%. ^1^H-NMR (CDCl_3_) δ: 1.52 (s, 9H, t-Bu), 4.85 (s, 2H, CH_2_), 7.31–7.33 (m, 2H), 7.35–7.38 (m, 1H), 7.40–7.44 (m, 2H), 7.47–7.49 (m, 2H), 7.54–7.57 (m, 1H), 7.87–7.89 (m, 2H) (2C_6_H_5_), ^13^C-NMR (CDCl_3_) δ: 27.8, 47.7, 55.3, 126.5, 126.9, 128.8, 128.9, 129.0, 131.7, 132.6, 141.1, 153.0, 163.5; MS (ESI^−^) *m*/*z* 370.4 (M-H)^−^.

N-[1-*tert*-Butyl-2-oxo-3-phenylimidazolidin-4-ylidene]-4-methylbenzenesulfonamide (**9c**): the title compound was purified by crystallization from hot EtOH. White solid. Yield 320 mg; 40%. ^1^H-NMR (CDCl_3_) δ: 1.52 (s, 9H, t-Bu), 2.41 (s, 3H, CH_3_), 4.83 (s, 2H, CH_2_), 7.26–7.28 (m, 2H), 7.30–7.32 (m, 2H), 7.34–7.37 (m, 2H), 7.40–7.43 (m, 1H), 7.75–7.77 (m, 2H) (C_6_H_5_ + C_6_H_4_), ^13^C-NMR (CDCl_3_) δ: 21.5, 27.8, 47.6, 55.3, 126.6, 126.9, 128.6, 128.8, 129.4, 131.7, 138.3, 143.4, 153.1, 163.3; MS (ESI^−^) *m*/*z* 384.3 (M-H)^−^.

4-Bromo-N-[1-*tert*-butyl-2-oxo-3-phenylimidazolidin-4-ylidene]benzenesulfonamide (**9d**): the title compound was purified by crystallization from hot EtOH. White solid. Yield 320 mg; 34%. ^1^H-NMR (CDCl_3_) δ: 1.52 (s, 9H, t-Bu), 4.83 (s, 2H, CH_2_), 7.29–7.31 (m, 2H), 7.36–7.39 (m, 1H), 7.41–7.44 (m, 2H) (C_6_H_5_), 7.60–7.62 (m, 2H), 7.72–7.74 (m, 2H) (C_6_H_4_), ^13^C-NMR (CDCl_3_) δ: 27.8, 47.7, 55.4, 126.9, 127.6, 128.2, 128.8, 128.9, 131.6, 132.1, 140.2, 152.9, 163.7; MS (ESI^−^) *m*/*z* 448.3/450.4 (M-H)^−^.

3-Bromo-N-[1-*tert*-butyl-2-oxo-3-phenylimidazolidin-4-ylidene]benzenesulfonamide (**9e**): the compound was purified by crystallization from hot EtOH. White solid. Yield 415 mg; 44%. ^1^H-NMR (CDCl_3_) δ: 1.53 (s, 9H, t-Bu), 4.83 (s, 2H, CH_2_), 7.30–7.39 (m, 4H), 7.42–7.45 (m, 2H), 7.66–7.68 (m,1H), 7.79–7.81 (m, 1H) 8.02 (m, 1H), (C_6_H_5_ + C_6_H_4_), ^13^C-NMR (CDCl_3_) δ: 27.8, 47.7, 55.4, 122.6, 125.1, 126.9, 128.8, 128.9, 129.6, 130.4, 131.6, 136.6, 142.9, 152.8, 163.9; MS (ESI^−^) *m*/*z* 448.4/450.4 (M-H)^−^.

N-[1-*tert*-Butyl-2-oxo-3-phenylimidazolidin-4-ylidene]-2,4,6-trimethylbenzenesulfonamide **(9f)**: the compound was purified by crystallization from hot EtOH. White solid. Yield 530 mg; 61%. ^1^H-NMR (CDCl_3_) δ: 1.52 (s, 9H, t-Bu), 2.27 (s, 3H, CH_3_), 2.58 (s, 6H, 2CH_3_), 4.82 (s, 2H, CH_2_), 6.89 (m, 2H, C_6_H_2_), 7.30–7.32 (m, 2H), 7.34–7.36 (m, 1H), 7.39–7.43 (m, 2H) (C_6_H_5_), ^13^C-NMR (CDCl_3_) δ: 20.9, 22.6, 27.8, 47.5, 55.2, 127.1, 128.6, 128.8, 131.5, 131.8, 135.1, 138.8, 142.1, 153.3, 162.9; MS (ESI^−^) *m*/*z* 412.4 (M-H)^−^.

N-[1-*tert*-Butyl-2-oxo-3-phenylimidazolidin-4-ylidene]naphthalene-1-sulfonamide (**9g**): the compound was purified by crystallization from hot EtOH/EtOAc mixture. White solid. Yield 540 mg; 61%. ^1^H-NMR (CDCl_3_) δ: 1.50 (s, 9H, t-Bu), 4.83 (s, 2H, CH_2_), 7.24–7.27 (m, 2H), 7.31–7.35 (m, 3H), (C_6_H_5_), 7.51–7.54 (m, 1H), 7.56–7.61 (m, 2H), 7.89–7.92 (m, 1H), 8.05–8.06 (m, 1H), 8.27–8.29 (m, 1H), 8.60–8.62 (m, 1H) (1-Naf), ^13^C-NMR (CDCl_3_) δ: 27.8, 47.6, 55.3, 123.9, 125.9, 126.8, 127.4, 127.7, 128.3, 128.5, 128.6, 128.7, 131.6, 134.2, 134.3, 136.3, 152.9, 163.7; MS (ESI^−^) *m*/*z* 420.4 (M-H)^−^.

N-[1-*tert*-Butyl-2-oxo-3-phenylimidazolidin-4-ylidene]naphthalene-2-sulfonamide (**9h**): the title compound was purified by crystallization from hot EtOH. White solid. Yield 175 mg; 20%. ^1^H-NMR (CDCl_3_) δ: 1.54 (s, 9H, t-Bu), 4.90 (s, 2H, CH_2_), 7.31–7.36 (m, 3H), 7.40–7.42 (m, 2H), (C_6_H_5_), 7.58–7.65 (m, 2H), 7.83–7.95 (m, 4H), 8.46 (m, 1H) (2-Naf), ^13^C-NMR (CDCl_3_) δ: 27.8, 47.7, 55.3, 122.3, 126.9, 127.4, 127.5, 127.8, 128.7, 128.7, 128.9, 129.1, 129.3, 131.7, 131.9, 134.8, 138.0, 153.0, 163.5; MS (ESI^−^) *m*/*z* 420.4 (M-H)^−^.

#### 3.1.3. General Synthetic Procedure for N-[2-Oxoimidazolidin-4-ylidene]sulphonamides (**10a**–**h**–**12a**–**h**)

To a solution of corresponding 4-iminoimidazolidin-2-one (**8b-d**) (1.43 mmol) in dry CH_2_Cl_2_ (20 mL), placed in an ice–salt bath, dry pyridine (0.23 mL, 2.86 mmol) was added, followed by sulfonyl chloride (1.8 mmol). After 15 min, the ice bath was removed, and the reaction was stirred at r.t. until completed (TLC control). Then, the reaction mixture was diluted with CH_2_Cl_2_ (50 mL), and the organic phase was washed with HCl 1N sol. (20 mL), H_2_O (20 mL), NaHCO_3_ sat. sol. (20 mL) and brine. The organic phase was dried, and the solvent evaporated under reduced pressure. The obtained product was purified as described.

N-[1-*tert*-Butyl-3-cyclopropylmethyl-2-oxoimidazolidin-4-ylidene]methanesulfonamide (**10a**): the title compound was purified by flash chromatography (eluent: PE/acetone = 9/1 *v/v*). An analytically pure sample was obtained by crystallization from hexane to give a white solid. Yield: 100 mg; 24%. M.p. 83.5–84.0 °C (C_6_H_14_). ^1^H-NMR (DMSO-d6) δ: 0.29–0.31 (m, 2H, CH_2_), 0.44–0.46 (m, 2H, CH_2_), 1.09 (m, 1H, CH) (Cp), 1.37 (s, 9H, t-Bu), 3.06 (s, 3H, SO_2_CH_3_), 3.31 (d, J^3^_HH_ = 7.2 Hz; 2H, CH_2_), 4.68 (s, 2H, CH_2_); ^13^C-NMR (DMSO-d6) δ: 3.8, 9.2, 27.8, 42.4, 44.9, 47.4, 54.8, 154.0, 164.3. MS (ESI^−^) *m*/*z* 286.3 (M-H)^−^.

N-[1-*tert*-Butyl-3-cyclopropylmethyl-2-oxoimidazolidin-4-ylidene]benzenesulfonamide (**10b**): the obtained colorless oil was solidified by treating with cold PE. The compound was further purified by crystallization from hot hexane to give a white solid. Yield: 200 mg; 40%. M.p. 114.5–115.0 °C (C_6_H_14_). ^1^H-NMR (CDCl_3_) δ: 0.26–0.29 (m, 2H, CH_2_), 0.41–0.44 (m, 2H, CH_2_), 1.10–1.12 (m, 1H, CH) (Cp), 1.46 (s, 9H, t-Bu), 3.40 (d, J^3^_HH_ = 7.2 Hz; 2H, CH_2_), 4.65 (s, 2H, CH_2_), 7.51–7.53 (m, 2H), 7.58–7.60 (m, 1H), 7.94–7.96 (m, 2H) (C_6_H_5_); ^13^C-NMR(CDCl_3_) δ: 3.8, 9.2, 27.8, 45.0, 47.8, 54.8, 126.5, 128.8, 132.5, 141.4, 153.9, 164.3. MS (ESI^−^) *m*/*z* 348.5 (M-H)^−^.

N-[1-*tert*-Butyl-3-cyclopropylmethyl-2-oxoimidazolidin-4-ylidene]-4-methylbenzenesulfonamide (**10c**): the obtained oil was purified by flash chromatography (eluent PE/acetone 9/1 *v*/*v*) and the resulting solid was purified further by crystallization from hot hexane. White solid. Yield: 135 mg; 26%. M.p. 104.5–105.0 °C (C_6_H_14_). ^1^H-NMR (CDCl_3_) δ: 0.27–0.30 (m, 2H, CH_2_), 0.42–0.44 (m, 2H, CH_2_), 1.10–1.13 (m, 1H, CH) (Cp), 1.46 (s, 9H, t-Bu), 2.44 (s, 3H, CH_3_Ph), 3.40 (d, J^3^_HH_ = 7.2 Hz; 2H, CH_2_Cp), 4.65 (s, 2H, CH_2_), 7.31–7.33 (m, 2H), 7.82–7.84 (m, 2H) (C_6_H_4_); ^13^C-NMR (CDCl_3_) δ: 3.8, 9.3, 21.6, 27.8, 45.0, 47.7, 54.8, 126.6, 129.4, 138.6, 143.3, 154.0, 164.2. MS (ESI^−^) *m*/*z*: 362.4 (M-H)^−^.

4-Bromo-N-[1-*tert*-Butyl-3-cyclopropylmethyl-2-oxoimidazolidin-4-ylidene]benzenesulfonamide (**10d**): the obtained oil was purified by flash chromatography (eluent PE/acetone 9/1), and the resulting solid was purified further by crystallization from hot hexane. White solid. Yield: 130 mg; 21%. M.p. 98.0–98.5 °C (C_6_H_14_). ^1^H-NMR (CDCl_3_) δ: 0.26–0.28 (m, 2H, CH_2_), 0.43–0.45(m, 2H, CH_2_), 1.08–1.11 (m, 1H, CH) (Cp), 1.46 (s, 9H, t-Bu), 3.40 (d, J^3^_HH_ = 7.2 Hz; 2H, CH_2_), 4.64 (s, 2H, CH_2_), 7.65–7.67 (m, 2H), 7.80–7.82 (m, 2H) (C_6_H_4_); ^13^C-NMR (CDCl_3_) δ: 3.8, 9.2, 27.8, 45.1, 47.8, 54.9, 127.4, 128.1, 132.1, 140.5, 153.8, 164.5. MS (ESI^−^) *m*/*z* 426.2/428.3 (M-H)^−^.

3-Bromo-N-[1-*tert*-Butyl-3-cyclopropylmethyl-2-oxoimidazolidin-4-ylidene]benzenesulfonamide (**10e**): the obtained oil was purified by flash chromatography (eluent PE/acetone 9/1) and the resulting solid was purified further by crystallization from hot hexane. White solid. Yield: 285 mg; 47%. M.p. 114.0–115.0 °C (C_6_H_14_). ^1^H-NMR (CDCl_3_) δ: 0.28–0.30 (m, 2H, CH_2_), 0.45–0.47 (m, 2H, CH_2_), 1.09–1.12 (m, 1H, CH) (Cp), 1.47 (s, 9H, t-Bu), 3.41 (d, J^3^_HH_ = 7.2 Hz; 2H, CH_2_), 4.65 (s, 2H, CH_2_), 7.39–7.42 (m, 1H), 7.71–7.72 (m, 1H), 7.88–7.90 (m, 1H), 8.09 (m, 1H) (C_6_H_4_); ^13^C-NMR (CDCl_3_) δ: 3.8, 9.2, 27.8, 45.2, 47.9, 54.9, 122.7, 125.1, 129.6, 130.4, 135.6, 143.2, 153.8, 164.7. MS (ESI^−^) *m*/*z* 426.4/428.3 (M-H)^−^.

N-[1-*tert*-Butyl-3-cyclopropylmethyl-2-oxoimidazolidin-4-ylidene]-2,4,6-trimethylbenzenesulfonamide (**10f**): the obtained oil was purified by flash chromatography (eluent PE/acetone 95/5 *v*/*v*) and the resulting solid was purified further by crystallization from hot hexane. White solid. Yield: 200 mg; 36%. M.p. 149.0–150.0 °C (C_6_H_14_). ^1^H-NMR (CDCl_3_) δ: 0.27–0.30 (m, 2H, CH_2_), 0.44–0.46 (m, 2H, CH_2_), 1.11–1.14 (m, 1H, CH) (Cp), 1.46 (s, 9H, t-Bu), 2.31 (s, 3H, CH_3_Ar), 2.68 (s, 6H, 2CH_3_Ar), 3.38 (d, J^3^_HH_ = 7.2 Hz; 2H, CH_2_), 4.62 (s, 2H, CH_2_), 6.96 (s, 2H) (C_6_H_2_); ^13^C-NMR (CDCl_3_) δ: 3.8, 9.3, 20.9, 22.7, 27.8, 44.9, 47.5, 54.7, 131.6, 135.4, 138.7, 142.0, 154.2, 163.8. MS (ESI^−^) *m*/*z* 390.5 (M-H)^−^.

N-[1-*tert*-Butyl-3-cyclopropylmethyl-2-oxoimidazolidin-4-ylidene]naphthalene-1-sulfonamide (**10g**): the obtained oil was purified by flash chromatography (eluent PE/acetone 9/1 *v*/*v*) and the resulting solid was purified further by crystallization from hot hexane. White solid. Yield: 250 mg; 44%. ^1^H-NMR (CDCl_3_) δ: 0.19–0.21 (m, 2H, CH_2_), 0.34–0.36 (m, 2H, CH_2_), 1.06–1.09 (m, 1H, CH) (Cp), 1.46 (s, 9H, t-Bu), 3.36 (d, J^3^_HH_ = 7.2 Hz; 2H, CH_2_), 4.68 (s, 2H, CH_2_), 7.55–7.68 (m, 3H), 7.93–7.95 (m, 1H), 8.08–8.09 (m, 1H), 8.31–8.32 (m, 1H), 8.75–8.77 (m, 1H) (C_10_H_7_); ^13^C-NMR (CDCl_3_) δ: 3.8, 9.2, 27.8, 45.1, 47.6, 54.8, 124.0, 125.8, 126.8, 127.4, 127.8, 128.5, 128.6, 134.1, 134.2, 136.6, 154.0, 164.7. MS (ESI^−^) *m*/*z* 398.5 (M-H)^−^.

N-[1-*tert*-Butyl-3-cyclopropylmethyl-2-oxoimidazolidin-4-ylidene]naphthalene-2-sulfonamide (**10h**): the obtained foam was partially purified by flash chromatography (eluent PE/EtOAc 9/1 *v*/*v*) and further by HPLC RP-18 (eluent CH_3_CN/H_2_O 7/3). Finally, an analytically pure sample was obtained by crystallization from hot EtOAc. White solid. Yield: 150 mg; 26%. M.p. 145.5–146.0 °C (EtOAc). ^1^H-NMR (DMSO-d6) δ: 0.17–0.19 (m, 2H, CH_2_), 0.31–0.33 (m, 2H, CH_2_), 0.95 (m, 1H, CH) (Cp), 1.40 (s, 9H, t-Bu), 3.30 (d, J^3^_HH_ = 7.2 Hz; 2H, CH_2_), 4.83 (s, 2H, CH_2_), 7.68–7.71 (m, 2H), 7.91–7.93 (m, 1H), 8.04–8.06 (m, 1H), 8.12–8.13 (m, 1H), 8.18–8.20 (m, 1H), 8.55–8.56 (m, 1H) (C_10_H_7_); ^13^C-NMR (DMSO-d6) δ: 3.5, 9.2, 27.1, 44.1, 48.11, 54.1, 122.33, 126.7, 127.6, 127.6, 129.2, 129.3, 129.4, 138.72, 153.3, 165.6. MS (ESI^−^) *m*/*z* 398.4 (M-H)^−^.

N-[1-*tert*-Butyl-3-(2-methoxyethyl)-2-oxoimidazolidin-4-ylidene]methanesulfonamide (**11a**): the product was purified by flash chromatography (eluent PE/acetone 9/1 *v*/*v*). The obtained colorless oil solidified upon standing in the desiccator. White solid. Yield: 130 mg; 31%. M.p. 70.5–71.0 °C. ^1^H-NMR (DMSO-d6) δ: 1.37 (s, 9H, t-Bu), 3.06 (s, 3H, CH_3_O), 3.24 (s, 3H, CH_3_SO_2_), 3.50 (t, 2H, CH_2_), 3.63 (t, 2H, CH_2_) 4.65 (s, 2H, CH_2_); ^13^C-NMR (DMSO-d6) δ:27.1, 41.7, 47.4, 54.0, 57.8, 67.3, 153.4, 165.6. MS (ESI^−^) *m*/*z* 290.3 (M-H)^−^.

N-[1-tert-Butyl-3-(2-methoxyethyl)-2-oxoimidazolidin-4-ylidene]benzenesulfonamide **(11b)**: the compound was purified by flash chromatography (eluent EP/acetone 9/1 *v*/*v*) and subsequently crystallized from hot i-Pr_2_O. White solid. Yield: 120 mg; 24%. M.p. 104.5–105.0 °C (i-Pr_2_O). ^1^H-NMR (DMSO-d6) δ: 1.38 (s, 9H, t-Bu), 3.10 (s, 3H, CH_3_O), 3.42 (t, 2H, CH_2_), 3.60 (t, 2H, CH_2_) 4.75 (s, 2H, CH_2_), 7.58–7.66 (m, 3H), 7.90–7.91 (m, 2H) (C_6_H_5_); ^13^C-NMR (DMSO-d6) δ: 27.1, 47.9, 54.1, 57.6, 67.4, 126.1, 129.1, 132.7, 141.7, 153.1, 165.7. MS (ESI^−^) *m*/*z* 352.4 (M-H)^−^.

N-[1-*tert*-Butyl-3-(2-methoxyethyl)-2-oxoimidazolidin-4-ylidene]-4-methylbenzene-1-sulfonamide **(11c)**: the compound was purified by flash chromatography (eluent PE/acetone 9/1 *v*/*v*). The obtained colorless oil solidified upon standing. Yield: 315 mg; 60%. M.p. 106.0–107.0 °C. ^1^H-NMR (DMSO-d6) δ: 1.38 (s, 9H, t-Bu), 2.39 (s, 3H, CH_3_Ar), 3.11 (s, 3H, CH_3_O), 3.41 (t, J^3^_HH_ = 5.9 Hz; 2H, CH_2_), 3.59 (t, J^3^_HH_ = 5.9 Hz; 2H, CH_2_) 4.73 (s, 2H, CH_2_), 7.38–7.40 (m, 2H), 7.77–7.79 (m, 2H) (C_6_H_4_); ^13^C-NMR (DMSO-d6) δ: 21.0, 27.1, 47.9, 54.1, 57.6, 67.4, 126.2, 129.5, 138.9, 142.9, 153.1, 165.4. MS (ESI^−^) *m*/*z* 366.4 (M-H)^−^.

4-Bromo-N-[1-*tert*-butyl-3-(2-methoxyethyl)-2-oxoimidazolidin-4-ylidene]benzenesulfonamide (**11d**): the compound was purified by flash chromatography (eluent PE/acetone 9/1 *v*/*v*) to give a white solid. Yield: 190 mg; 31%. M.p. 103.5–104.0 °C. ^1^H-NMR (DMSO-d6) δ: 1.38 (s, 9H, t-Bu), 3.12 (s, 3H, CH_3_O), 3.42 (t, J^3^_HH_ = 5.9 Hz; 2H, CH_2_), 3.60 (t, J^3^_HH_ = 5.9 Hz; 2H, CH_2_) 4.75 (s, 2H, CH_2_), 7.80–7.84 (m, 4H, C_6_H_4_); ^13^C-NMR (DMSO-d6) δ: 27.1, 48.0, 54.1, 57.6, 67.4, 126.4, 128.3, 132.1, 141.0, 153.0, 166.1. MS (ESI^−^) *m*/*z* 430.4/432.2 (M-H)^−^.

3-Bromo-N-[1-*tert*-butyl-3-(2-methoxyethyl)-2-oxoimidazolidin-4-ylidene]benzenesulfonamide (**11e**): the compound was purified by flash chromatography (eluent PE/acetone 9/1 *v*/*v*). The obtained colorless oil solidified upon standing in the desiccator. Yield: 355 mg; 57%. M.p. 104.0–105.0 °C. ^1^H-NMR (DMSO-d6) δ: 1.38 (s, 9H, t-Bu), 3.13 (s, 3H, CH_3_O), 3.43 (t, J^3^_HH_ = 5.9 Hz; 2H, CH_2_), 3.61 (t, J^3^_HH_ = 5.9 Hz; 2H, CH_2_) 4.77 (s, 2H, CH_2_), 7.55–7.57 (m, 1H), 7.86–7.91 (m, 2H), 8.03 (m, 1H) (C_6_H_4_). ^13^C-NMR (DMSO-d6) δ: 27.6, 48.7, 54.7, 58.2, 68.0, 122.5, 125.7, 129.1, 131.9, 136.0, 144.3, 153.5, 166.9. MS (ESI-) *m*/*z* 430.3/432.4 (M-H)^−^.

N-[1-*tert*-Butyl-3-(2-methoxyethyl)-2-oxoimidazolidin-4-ylidene]-2,4,6-trimethylbenzene-1-sulfonamide (**11f**): the compound was purified by flash chromatography (eluent PE/acetone 9/1) to give a white solid. Yield: 350 mg; 62%. M.p. 141.0–143.5 ° C. ^1^H-NMR (DMSO-d6) δ: 1.37 (s, 9H, t-Bu), 2.26 (s, 3H, CH_3_Ar), 2.58 (s, 6H, 2CH_3_Ar), 3.14 (s, 3H, CH_3_O), 3.43 (m, 2H, CH_2_), 3.58 (t, J^3^_HH_ = 5.9 Hz; 2H, CH_2_), 4.63 (s, 2H, CH_2_), 7.04 (s, 2H, C_6_H_2_); ^13^C-NMR (DMSO-d6) δ: 20.5, 22.2, 27.1, 47.4, 54.1, 57.7, 67.5, 131.5, 135.7, 138.0, 141.7, 153.3, 164.8. MS (ESI^−^) *m*/*z* 394.4 (M-H)^−^.

N-[1-*tert*-Butyl-3-(2-methoxyethyl)-2-oxoimidazolidin-4-ylidene]naphthalene-1-sulfonamide (**11g**): the compound was purified by flash chromatography (eluent EdP/acetone 9/1 *v*/*v*) and subsequently crystallized from hot EtOH. White solid. Yield: 145 mg; 25%. M.p. 110.5–111.0 °C (EtOH). ^1^H-NMR (DMSO-d6) δ: 1.37 (s, 9H, t-Bu), 2.97 (s, 3H, CH_3_O), 3.34 (t, 2H, CH_2_), 3.56 (t, 2H, CH_2_) 4.73 (s, 2H, CH_2_), 7.66–7.74 (m, 3H), 8.09 (m, 1H), 8.23–8.26 (m, 2H), 8.65–8.67 (m, 1H) (C_10_H_7_); ^13^C-NMR (DMSO-d6) δ: 27.1, 47.7, 54.2, 57.5, 67.2, 124.5, 125.5, 126.9, 127.2, 127.8, 128.8, 133.8, 134.0, 136.7, 153.1, 156.7, 166.0. MS (ESI^−^) *m*/*z* 402.4 (M-H)^−^.

N-[1-*tert*-Butyl-3-(2-methoxyethyl)-2-oxoimidazolidin-4-ylidene]naphthalene-2-sulfonamide **(11h)**: the obtained white solid was purified by crystallization from hot i-Pr_2_O. Yield: 245 mg; 42%. M.p. 129.5–130.5 °C (i-Pr_2_O). ^1^H-NMR (DMSO-d6) δ: 1.39 (s, 9H, t-Bu), 3.08 (s, 3H, CH_3_O), 3.42 (t, 2H, CH_2_), 3.61 (t, 2H, CH_2_) 4.81 (s, 2H, CH_2_), 7.68–7.72 (m, 2H), 7.91–7.93 (m, 1H), 8.05–8.06 (m, 1H), 8.12–8.13 (m, 1H), 8.18–8.19 (m, 1H), 8.56 (m, 1H) (C_10_H_7_); ^13^C-NMR (DMSO-d6) δ: 27.1, 48.0, 54.1, 57.6, 67.4, 126.3, 126.6, 127.6, 127.8, 128.8, 129.2, 129.3, 131.7, 134.2, 138.8, 153.1, 165.8. MS (ESI^−^) *m*/*z* 402.4 (M-H)^−^.

N-[1-Adamant-1-yl-3-cyclopropylmethyl-2-oxoimidazolidin-4-ylidene]methanesulfonamide (**12a**): the title compound was purified by crystallization from hot MeOH. White solid. Yield: 105 mg; 20%. M.p. 199.5–200.0 °C (MeOH). ^1^H-NMR (DMSO-d6) δ: 0.28–0.30 (m, 2H, CH_2_), 0.44–0.46 (m, 2H, CH_2_), 1.07–1.10 (m, 1H, CH) (Cp), 1.63 (m, 6H, 3CH_2_), 2.07 (m, 9H, 3CH_2_ + 3CH) (Ad), 3.06 (s, 3H, CH_3_SO_2_), 3.30 (d, J^3^_HH_ = 7.2 Hz; 2H, CH_2_), 4.67 (s, 2H, CH_2_); ^13^C-NMR (DMSO-d6) δ: 3.6, 9.2, 28.8, 35.6, 41.7, 43.9, 46.5, 54.6, 153.2, 165.7. MS (ESI^−^) *m*/*z* 364.3 (M-H)^−^.

N-[1-Adamant-1-yl-3-cyclopropylmethyl-2-oxoimidazolidin-4-ylidene]benzenesulfonamide (**12b**): the title compound was purified by crystallization from hot MeOH. White solid. Yield: 275 mg; 45%. M.p. 198.0–198.5 °C (MeOH). ^1^H-NMR (DMSO-d6) δ: 0.18–0.20 (m, 2H, CH_2_), 0.34–0.36 (m, 2H, CH_2_), 0.94 (m, 1H, CH) (Cp), 1.64 (m, 6H, 3CH_2_), 2.09 (m, 9H, 3CH_2_ + 3CH) (Ad), 3.28 (d, J^3^_HH_ = 7.2 Hz; 2H, CH_2_), 4.77 (s, 2H, CH_2_), 7.58–7.66 (m, 3H), 7.90–7.91 (m, 2H) (C_6_H_5_); ^13^C-NMR (DMSO-d6) δ: 3.4, 9.1, 28.9, 35.6, 38.9, 44.0, 47.1, 54.7, 126.2, 129.1, 132.6, 141.7, 152.9. MS (ESI^−^) *m*/*z* 426.4 (M-H)^−^.

N-[1-Adamant-1-yl-3-cyclopropylmethyl-2-oxoimidazolidin-4-ylidene]-4-methyl-1-benzenesulfonamide (**12c**): the title compound was purified by crystallization from hot MeOH. White solid. Yield: 220 mg; 35%. M.p. 188.5–189.0 °C (MeOH). ^1^H-NMR (DMSO-d6) δ: 0.19–0.20 (m, 2H, CH_2_), 0.34–0.36 (m, 2H, CH_2_), 0.93–0.96 (m, 1H, CH) (Cp), 1.64 (m, 6H, 3CH_2_), 2.08 (m, 9H, 3CH_2_ + 3CH) (Ad), 2.39 (s, 3H, CH_3_Ar), 3.27 (d, J^3^_HH_ = 7.2 Hz; 2H, CH_2_), 4.75 (s, 2H, CH_2_), 7.38–7.40 (m, 2H), 7.77–7.79 (m, 2H) (C_6_H_4_); ^13^C-NMR (DMSO-d6) δ: 3.5, 9.2, 21.0, 28.9, 35.6, 38.9, 44.0, 47.0, 54.7, 126.3, 129.5, 138.9, 143.0, 153.0, 165.5. MS (ESI^−^) *m*/*z* 440.4 (M-H)^−^

N-[1-Adamant-1-yl-3-cyclopropylmethyl-2-oxoimidazolidin-4-ylidene]-4-bromo-1-benzenesulfonamide (**12d**): the obtained white foam was purified by flash chromatography (eluent PE/acetone 95/5 *v*/*v*) and then crystallized from hot hexane. White solid. Yield: 290 mg; 40%. M.p 176.0–176.5 °C (C_6_H_14_). ^1^H-NMR (DMSO-d6) δ: 0.19–0.21 (m, 2H, CH_2_), 0.36–0.37 (m, 2H, CH_2_), 0.94–0.96 (m, 1H, CH) (Cp), 1.64 (m, 6H, 3CH_2_), 2.08 (m, 9H, 3CH_2_ + 3CH) (Ad), 3.28 (d, J^3^_HH_ = 7.2 Hz; 2H, CH_2_), 4.76 (s, CH_2_), 7.79–7.85 (m, 4H, C_6_H_4_); ^13^C-NMR (DMSO-d6) δ: 3.5, 9.1, 28.9, 35.6, 38.9, 44.1, 47.2, 54.8, 126.4, 128.3, 132.2, 141.0, 152.8, 166.1. MS (ESI^−^) *m*/*z* 504.4/506.3 (M-H)^−^.

N-[1-Adamant-1-yl-3-cyclopropylmethyl-2-oxoimidazolidin-4-ylidene]-3-bromo-1-benzenesulfonamide (**12e**): the obtained yellow oil was purified by flash chromatography (eluent PE/acetone 95/5 *v*/*v*) to give the title compound as a white solid. Yield: 390 mg; 54%. M.p. 138.0–138.5 °C. ^1^H-NMR (DMSO-d6) δ: 0.19–0.21 (m, 2H, CH_2_), 0.35–0.37 (m, 2H, CH_2_), 0.93–0.96 (m, 1H, CH) (Cp), 1.64 (m, 6H, 3CH_2_), 2.09 (m, 9H, 3CH_2_ + 3CH) (Ad), 3.28 (d, J^3^_HH_ = 7.2 Hz; 2H, CH_2_), 4.79 (s, 2H, CH_2_), 7.55–7.56 (m, 1H), 7.86–7.93 (m, 2H), 8.03–8.04 (m, 1H) (C_6_H_4_); ^13^C-NMR (DMSO-d6) δ: 3.5, 9.2, 28.9, 35.6, 38.9, 44.1, 47.3, 54.8, 122.0, 125.3, 128.6, 131.4, 135.5, 143.7, 152.8, 166.4. MS (ESI^−^) *m*/*z* 504.4/506.3 (M-H)^−^.

N-[1-Adamant-1-yl-3-cyclopropylmethyl-2-oxoimidazolidin-4-ylidene]-2,4,6-trimethyl-1-benzenesulfonamide (**12f**): the obtained yellow oil obtained was purified by flash chromatography (eluent PE/acetone 95/5 *v*/*v*) to give the title compound as a white solid. Yield: 180 mg; 27%. M.p. 166.5–167.5 °C. ^1^H-NMR (DMSO-d6) δ: 0.20–0.21 (m, 2H, CH_2_), 0.38–0.39 (m, 2H, CH_2_), 0.96–0.98 (m, 1H, CH) (Cp), 1.63 (m, 6H, 3CH_2_), 2.07 (m, 9H, 3CH_2_ + 3CH) (Ad), 2.26 (s, 3H, CH_3_Ar), 2.58 (s, 6H, 2CH_3_Ar), 3.26 (d, J^3^_HH_ = 7.2 Hz; 2H, CH_2_), 4.66 (s, 2H, CH_2_), 7.04 (s, 2H, C_6_H_2_); ^13^C-NMR (DMSO-d6) δ: 3.5, 9.2, 20.5, 22.3, 28.9, 35.6, 38.9, 44.0, 46.6, 54.7, 131.5, 135.7, 137.9, 141.7, 153.1, 164.9. MS (ESI^−^) *m*/*z*: 468.6 (M-H)^−^.

N-[1-Adamant-1-yl-3-cyclopropylmethyl-2-oxoimidazolidin-4-ylidene]naphtalene-1-sulfonamide (**12g**): the title compound was purified by crystallization from hot MeOH. White solid. Yield: 165 mg; 24%. M.p. 188.5–189.0 °C (MeOH). ^1^H-NMR (DMSO-d6) δ: 0.08–0.11 (m, 2H, CH_2_), 0.23–0.26 (m, 2H, CH_2_), 0.88–0.92 (m, 1H, CH) (Cp), 1.63 (m, 6H, 3CH_2_), 2.07 (m, 9H, 3CH_2_ + 3CH) (Ad), 3.23 (d, J^3^_HH_ = 7.2 Hz; 2H, CH_2_), 4.75 (s, 2H, CH_2_), 7.65–7.74 (m, 3H), 8.08–8.09 (m, 1H), 8.22–8.26 (m, 2H), 8.65–8.67 (m, 1H) (C_10_H_7_); ^13^C-NMR (DMSO-d6) δ: 3.4, 9.0, 28.9, 35.6, 38.9, 44.2, 46.8, 54.8, 124.5, 125.5, 126.9, 127.2, 127.7, 127.7, 128.8, 133.8, 134.0, 136.7, 152.9, 166.0. MS (ESI^−^) *m*/*z* 476.4 (M-H)^−^.

N-[1-Adamant-1-yl-3-cyclopropylmethyl-2-oxoimidazolidin-4-ylidene]naphtalene-2-sulfonamide (**12h**): the title compound was purified by crystallization from hot MeOH. White solid. Yield: 180 mg; 26%. M.p. 167.0–167.5 °C (MeOH). ^1^H-NMR (DMSO-d6) δ: 0.18–0.19 (m, 2H, CH_2_), 0.31–0.33 (m, 2H, CH_2_), 0.93–0.95 (m, 1H, CH) (Cp), 1.64 (m, 6H, 3CH_2_), 2.08–2.09 (m, 9H, 3CH_2_ + 3CH) (Ad), 3.28 (d, J^3^_HH_ = 7.2 Hz; 2H, CH_2_), 4.83 (s, 2H, CH_2_), 7.68–7.72 (m, 2H), 7.92–7.93 (m, 1H), 8.05–8.20 (m, 3H), 8.56 (m, 1H) (C_10_H_7_); ^13^C-NMR (DMSO-d6) δ: 4.0, 9.7, 29.4, 36.1, 44.6, 47.7, 55.3, 122.8, 127.2, 128.2, 128.3, 129.7, 129.9, 132.2, 134.7, 139.2, 153.4, 166.3. MS (ESI^−^) *m*/*z* 476.5 (M-H)^−^.

N-[3-(cyclopropylmethyl)-2-oxo-1-(propan-2-yl)imidazolidin-4-ylidene]naphthalene-1-sulfonamide (**13**): the compound was obtained following the same synthetic scheme, starting from i-PrNH_2_ (1.0 mL, 11.7 mmol). Intermediates were purified by flash chromatography and used without characterization. The title compound was purified by flash chromatography (eluent PE/acetone 9/1 *v*/*v*) and crystallized from hot EtOH. White solid. Overall yield 300 mg; 7%. ^1^H-NMR (DMSO-d6) δ: 0.11–0.12 (m, 2H, CH_2_), 0.25–0.27 (m, 2H, CH_2_), 0.91–0.93 (m, 1H, CH) (Cp), 1.17 (d, J^3^_HH_ = 6.9 Hz; 6H, 2CH_3_), 3.28 (d, J^3^_HH_ = 7.2 Hz; 2H, CH_2_), 4.05–4.08 (m, 1H, CH), 4.62 (s, 2H, CH_2_), 7.66–7.73 (m, 3H), 8.09–8.10 (m, 1H), 8.25–8.26 (m, 2H), 8.66–8.68 (m, 1H) (C_10_H_7_); ^13^C-NMR (DMSO-d6) δ: 3.4, 9.0, 19.7, 44.1, 44.5, 45.5, 124.5, 125.4, 126.9, 127.3, 127.6, 127.7, 128.8, 133.8, 134.0, 136.5, 153.6, 166.3. MS (ESI^−^) *m*/*z* 384.4 (M-H)^−^.

N-[1-*tert*-Butyl-3-(2-methylpropyl)-2-oxoimidazolidin-4-ylidene]-4-methylbenzene-1-sulfonamide (**14**): the compound was obtained following the same synthetic scheme, starting from 1-(2-methylpropyl)amine (1.5 mL, 15.1 mmol). Intermediates were purified by flash chromatography and used without characterization. The title compound was purified by flash chromatography (eluent PdE/EtOAc 9/1 *v*/*v*) to give a white solid. Overall yield 1.10 g; 20%. ^1^H-NMR (CDCl_3_) δ: 0.84 (d, J^3^_HH_ = 6.9 Hz; 6H, 2CH_3_), 1.45 (s, 9H, t-Bu), 2.03–2.06 (m, 1H, CH), 2.44 (s, 3H, ArCH_3_), 3.36 (d, J^3^_HH_ = 7.6 Hz; 2H, CH_2_), 4.63 (s, 2H, CH_2_), 7.31–7.32 (m, 2H), 7.81–7.83 (m, 2H) (C_6_H_4_); ^13^C-NMR (CDCl_3_) δ: 20.0, 21.5, 26.7, 27.7, 47.5, 47.6, 126.5, 129.4, 138.6, 143.3, 154.2, 166.4. MS (ESI^−^) *m*/*z* 364.4 (M-H)^−^.

N-[3-*tert*-Butyl-1-(cyclopropylmethyl)-2-oxoimidazolidin-4-ylidene]benzenesulfonamide (**15**): the compound was obtained following the same synthetic scheme, starting from cyclopropylmethylamine (0.60 mL, 6.9 mmol). Intermediates were purified by flash chromatography and used without characterization. The title compound was purified by flash chromatography (eluent CH_2_Cl_2_/acetone 99/1 *v*/*v*) to give a white solid. Overall yield 120 mg; 5%. ^1^H-NMR (DMSO-d6) δ: 0.21–0.22 (m, 2H, CH_2_), 0.48–0.50 (m, 2H, CH_2_), 0.95–0.97 (m, 1H, CH) (Cp), 1.52 (s, 9H, t-Bu), 3.14 (d, J^3^_HH_ = 7.2 Hz; 2H, CH_2_), 4.58 (s, 2H, CH_2_), 7.60–7.68 (m, 3H), 7.88–7.89 (m, 2H) (C_6_H_5_); ^13^C-NMR (DMSO-d6) δ: 3.1, 8.7, 28.0, 46.8, 48.1, 59.4, 125.8, 129.1, 132.5, 141.7, 154.6, 166.4. MS (ESI^−^) *m*/*z* 348.4 (M-H)^−^.

### 3.2. Molecular Docking Simulations

The structures of hCB2R in agonist and antagonist conformation were retrieved from the Protein Data Bank (PDB codes 6KPC [24] and 5ZTY [26], respectively) and used for molecular docking analyses with GOLD version 5.5 [35,36,37,38]. To verify the software capability of reproducing the co-crystallographic ligand binding poses, we extracted the cognate ligands (the AM12033 agonist in 6KPC and the full antagonist AM10257 in 5ZTY) and self-docked them to their corresponding binding site. We obtained RMSD values of 0.897 Å for self-docking in 6KPC and 1.05 Å for self-docking in 5ZTY.

The synthesized compounds were sketched with moldraw (Molecular Discovery Ltd.) and converted into the mol2 format using Open Babel [39]; their tautomeric/protonation state at physiological pH was checked using MoKa [40]. Compounds were first minimized using a combination of the steepest descent and conjugate gradient minimization methods and then submitted to molecular docking simulations. The region of interest was defined to contain all the residues within 10 Å of a reference atom (C_ζ_ of Phe183). No constraint was applied. The GOLD standard parameters were used, and the complex was subjected to 50 genetic algorithm runs. Finally, poses were scored with the CHEMPLP function and ranked accordingly.

### 3.3. In Vitro Pharmacological Evaluation

#### 3.3.1. Competition Binding Assay

Compound affinity for CB1R and CB2R was measured as previously reported [41]. Briefly, membranes from HEK-293 cells overexpressing the human recombinant CB1R (Bmax = 2.5 pmol/mg protein) and human recombinant CB2R (B_max_ = 4.7 pmol/mg protein) were incubated with [^3^H]-CP-55,940 (0.14 nM/Kd = 0.18 nM and 0.084 nM/K_d_ = 0.31 nM, respectively, for CB1R and CB2R) as the high-affinity ligand. Competition curves were performed by displacing [^3^H]-CP-55 940 with increasing concentrations of compounds (0.1 nM–10 or 25 μM). Nonspecific binding was defined by 10 μM WIN55 212−2 as the heterologous competitor (Ki values 9.2 and 2.1 nM, respectively, for CB1R and CB2R). IC_50_ values were determined for compounds showing >50% displacement at 10 μM. All compounds were tested following the procedure described by the manufacturer (Perkin-Elmer, Italy). Displacement curves were generated by incubating drugs with [^3^H]-CP-55 940 for 90 min at 30 °C. Ki values were calculated by applying the Cheng−Prusoff equation to the IC_50_ values for the displacement of the bound radioligand by increasing concentrations of the test compound. Data represent the mean values of three independent experiments performed in duplicate and are expressed as the average of K_i_ (µM) ± standard deviation. Data were analyzed using PRISM.9.3 software (GraphPad Software Inc, San Diego, CA, USA).

#### 3.3.2. Functional Activity at CB2R In Vitro

Gi-coupled cAMP modulation was measured following the manufacturer’s protocol (Eurofins, Fremont, CA, USA), as previously reported [29]. Briefly, CHO-K1 cells overexpressing the human CB2R were plated into a 96-well plate (10,000 cells/well) and incubated overnight at 37 °C, 5% CO_2_. Media was aspirated and replaced with 30 μL of assay buffer. Cells were incubated for 30 min at 37 °C with 15 μL of 3× dose−response solutions of samples prepared in the presence of a cell assay buffer containing 3× of 25 μM NKH-477 solution to stimulate adenylate cyclase and enhance basal cAMP levels. For those compounds not showing a decrease in cAMP levels, the effect upon receptor activation in the presence of the JWH-133 selective agonist was investigated. Cells were pre-incubated with samples (15 min at 37 °C at 6× the final desired concentration), followed by 30 min incubation with the JWH-133 agonist challenge at the EC_80_ concentration (EC_80_ = 4 μM, previously determined in separate experiments) in the presence of NKH-477 to stimulate adenylate cyclase and enhance cAMP levels. Cell lysis and cAMP detection were performed as per the manufacturer’s protocol. Luminescence measurements were performed using a GloMax Multi Detection System (Promega, Italy). Data are reported as the means ± SEM of three independent experiments conducted in triplicate and were normalized considering the NKH-477 stimulus alone as 100% of the response. Data were analyzed using PRISM.9.3 software (GraphPad Software Inc, San Diego, CA, USA).

## 4. Conclusions

We reported here the development and SAR of a new series of hCB2R modulators with a 2-oxoimidazolidin-4-ylidene sulfonamide scaffold.

The most active compounds showed a high selectivity towards hCB2R with respect to hCB1R and an inhibition constant in the low micromolar range. Interestingly many of the most active compounds showed a selective agonist effect, which was predicted by means of structure-based in silico simulations. By comparing the behavior and structure of our new compounds with other agonists and antagonists reported in the literature, we have been able to confirm the critical role played by Trp258 and the importance of the toggle switch mechanism in determining the compound agonist/antagonist effect. Accordingly, the nature of the R, R′ and R″ substituents, even if no polar contact is formed, appear to be essential for stabilizing the hCB2R agonist or antagonist form and, thus, the effect of this class of three-arm ligands. We had aimed to identify selective ligands of the hCB2 receptor and create a predictive in silico model for further scaffold optimization. Keeping in mind the relevance of tempering cannabinoid receptor signaling, the next step will be to explore the possibility of this type of ligand acting as an allosteric modulator, avoiding the inherent side effects of orthosteric ligands. Interestingly, structure-based simulations enabled the identification of six pure hCB2R-selective ligands that will be considered as starting points for the development of more potent ligands, allowing the easy and rapid identification of agonists with respect to antagonist compounds. This is an important achievement as the therapeutic potential of hCB2 antagonism/inverse agonism is yet to be elucidated. CB2-specific inverse agonists have been reported to ameliorate bone damage in a rat model of relapsing–remitting arthritis [42] and have shown anti-inflammatory and anti-osteoclastogenic properties in activated macrophages and differentiating osteoclasts, respectively [43]. Unveiling the therapeutic potential of antagonists/inverse agonists (in addition to full agonists) will provide critical clues for the rational design of compounds that can ameliorate the pathological conditions characterized by a hyperactive CB2 tone.

## Data Availability

Data supporting the findings of this study are available from the corresponding authors, K.C. and A.L., upon reasonable request.

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
