# Peer review of "N-[1,3-Dialkyl(aryl)-2-oxoimidazolidin-4-ylidene]-aryl(alkyl)sulphonamides as Novel Selective Human Cannabinoid Type 2 Receptor (hCB2R) Ligands; Insights into the Mechanism of Receptor Activation/Deactivation"

_molecules, 2022, doi:10.3390/molecules27238152_

Round 1

Reviewer 1 Report

In this manuscript, Chegaev and coworker reported a novel series of selective human cannabinoids type 2 receptors (hCB2R) ligands. Based on a hit molecule 1, a systematic SAR was performed on the substituents (R, R1 and R2) to improve the affinity. Consequently, they abled to find out a potent molecule 10g with a Ki of <100 nM against hCB2R ligand, while it is completely inactive towards hCB1R ligand even at 10 uM concentration. Later they developed an in-silico model to predict and validate functional activity. Considering the novelty and usefulness of this work in future drug discovery study, this manuscript should be considered for publication after addressing following comments.

1.      R’ and R’’ for compound in Figure 2 should be defined.

2.      R’ and R1 were interchangeably used in Scheme 3 and in the multiple instances in the main text. It should be consistent (similar comment for R’’ and R2).

3.      At least one reference of SR144528 should be added in the line 131.

4.      Competitive binding curve against hCB2R for all potent compounds (10d, 10e, 10f, 10g, 10h, 12a, 12b, 12e, 13) should be included in the manuscript or in the supporting information.

5.      Figure 3: Check the structure for compound 13.

6.      Spell check for “a week ligand” (line 77) and “aminoacetinitriles” (line 104),

7.      “cicloproylmethyl” should be changed into “cyclopropylmethyl”. There are multiple instances of this typographical error in between page 16 and 19.

8.      1H and 13C NMR spectra for all final compounds should be added to the supporting information.

9.      HZ value of 1H NMR and 13C NMR should be mentioned during presenting NMR data.

10.   Isolated amount for all final products and intermediates should be mentioned while reporting isolated yield.

11.   The scale of the reaction and amount of the product (compounds 1 and 4a-4g) should be mentioned during presenting synthetic procedures of them.

12.   1H NMR data and MS value for compound 13 should be checked.

Author Response

Authors thank reviewer for the valuable advises for improvement of the manuscript. All the requested corrections were performed.

Reviewer 2 Report

This is a very solid medicinal chemistry paper supported by molecular docking simulations, binding assays and some functional work. There are some suggestions for improvements.

1. The major concern is the somewhat limited nature of functional assays, which fail to match the strength of the preceding work. The inhibition of cAMP studies are fine, but the study would benefit from additional assays, in particular GTPgammaS turnover, which are quite standard for cannabinoid receptors.

2. Some potentially interesting further insights into the agonists (and also antagonist) identified are slightly lost with the decision to present a combined Results and Discussion rather than separate sections (it is unclear if this is journal policy). For example, the authors may want to speculate if ligands could act at allosteric sites and what is the clinical potential of these compounds, including if compound 12e a potential useful antagonist.

3. Related to above, some relevant literature on recent work on CB2 ligands appears to be missing, for example https://doi.org/10.1177/2472555217748403 

Author Response

  1. We thank the reviewer for the suggestions. We fully agree with her/him this type of studies would take benefit from investigating GPCR activation response at different levels of the signaling pathway. However, the scope of this paper was to identify selective agonists of hCB2 receptor and to create a predictive in-silico model for further optimization of our scaffold. We consider that at this point the inhibition of cAMP test is quite reliable and affordable and perfectly suit to our proposes. In further development of our ligands, we aim at improving the affinity maintaining the selectivity. Once more active compounds will be obtained more deepen studies will be performed.

  2. We edited the manuscript according to journal policy. Whitin the Conclusion, we added a few sentences to speculate about possible interactions at allosteric sites and we indicated pathological context where the use of CB2 antagonist were reported to be beneficial.
  3. Authors thank the reviewer for this indication, the paper reference was added in introduction section.

Reviewer 3 Report

The human cannabinoid type 2 receptor (hCB2R) is an important and novel drug target for several medical conditions such as pain and inflammation. The authors designed different N-[1,3-dialkyl(aryl)-2-oxoimidazolidin-4-ylidene]-aryl(alkyl)sulfonamides and studied their affinity for hCB2R (and hCB1R). They were able to identify the chemical features leading to finely tuned hCB2R selectivity. In addition, an in-silico model capable of predicting the functional activity of hCB2R ligands was proposed and validated. The proposed receptor activation/deactivation model enabled the identification of four pure hCB2R-selective agonists that can be used as a starting point for the development of more potent ligands.

General comments

The reviewer would like to congratulate the authors. This is great work and a big achievement! The clinical potential of this discovery should be more emphasized in the introduction. What are the specific disorders which would benefit from hCB2R activation treatment?

Minor comments

The reviewer would like to ask the authors to add some thoughts to the discussion about potential allergic (cross-)reactions to sulphonamides as potential limitations for the use of this class of substances. Also, are there medical conditions where hCB2R activation is detrimental?

Author Response

1. Author thank the reviewer for a congratulations. The scope of this paper was to identify selective agonists of hCB2 receptor and to create a predictive in-silico model for further optimization of our scaffold. Regarding the most promising compounds we identified, we consider too premature to discuss about their clinic potential. We considered adequate to have listed a number of pathologies were hCB2R are involved in the introduction section.

2. The activations of hCB2R plays a fundamental role in allergic inflammation like allergic dermatitis. However, controversial date have been reported. Some groups showed suppressed inflammatory edema and proliferation of matrix tissues (acanthosis) in CB2-KO mice exhibited compared with WT mice (Mimura et al., Int. Arch. Allergy Immunol. 2012). Other groups reported that CB2 activation could provoke exacerbated allergic reactions and edema in CB2-KO mice compared with WT mice (Karsac et al., Science 2007).

The allergic reaction on sulphonamides involve a considerable number of persons, but normally they are caused by antibacterial sulfonamides. Structurally, our compounds are completely different from sulphonamides (lack of acidic NH group and lack of nitrogen atom on para position of aromatic ring). Therefore, any speculations about their allergic properties would be unfounded. As we state before, at this point of our studies it is very premature to speak about drug-like properties and side effects of our compounds.

As far as reviewer query about when hCB2R activation can be considered detrimental, osteoporosis and perimenopause/menopause are medical conditions to be considered in this respect. A recent paper showed that a selective estrogen receptor (ER) modulator with reduced affinity for ERs exerted its anti-inflammatory and anti-osteoclastogenic properties via a selective affinity for CB2 acting as inverse agonist (Franks et al., Toxicol Appl Pharmacol. 2018). Nevertheless, we have to point out that all findings related to CB2 receptor modulation in bone metabolism are confounded by low study number and heterogenicity of models.